# Latent representation of H&E images retains biological information in a breast cancer cohort

Chloé Benmussa[1][¤][*], Esther Sanfeliu[2,3], Anabel Martínez-Romero[2,3], Blanca González-Farré[2,3], Tomás Pascual[2,4,5], Joaquín Gavilá[5,6], Alona Levy-Jurgenson[1,7], Ariel Shamir[1], Fara Brasó-Maristany[2,4,8], Aleix Prat[2,4,8,9,10], Zohar Yakhini[1,7]

**1** School of Computer Science, Reichman University IDC, Herzliya, Israel, **2** Translational Genomics and Targeted Therapies in Solid Tumors Group, August Pi i Sunyer Biomedical Research Institute (IDIBAPS), Barcelona, Spain, **3** Department of Pathology, Hospital Clínic of Barcelona, Barcelona, Spain, **4** Cancer Institute and Blood Diseases, Hospital Clínic of Barcelona, Barcelona, Spain, **5** SOLTI Breast Cancer Research Group, Barcelona, Spain, **6** Department of Medical Oncology, Instituto Valenciano de Oncología, Valencia, Spain, **7** Faculty of Computer Science, Technion, Haifa, Israel, **8** Reveal Genomics, Barcelona, Spain, **9** Department of Medicine, University of Barcelona, Barcelona, Spain, **10** Institute of Oncology (IOB)-Hospital QuirónSalud, Barcelona, Spain

¤ Current address: School of Computer Science, Reichman University IDC, Herzliya, Israel
* chloebenmussa@gmail.com

**Data availability statement:** Code is available at: https://github.com/ChloeBnms/Latent-representation-of-H-E-images. Data is

## Abstract

Imaging technologies and staining based pathology are important components of common practice cancer care. Specifically, H&E imaging is standard for almost all cancer patients. Traditionally, H&E images can serve, when used by experienced trained pathologists, to infer important biological properties of the samples. Recent work demonstrated that machine learning and machine vision analysis of H&E images can further expand the scope of the inference. However, H&E images are high-resolution, making them difficult to analyze and possibly noisy. In this work, we propose an autoencoder-based pipeline that greatly reduces the dimension of the data representation while maintaining valuable properties. In particular, we investigate how different latent space dimensions affect bulk label predictions from H&E. We use autoencoders applied to image tiles as a tool in this investigation and also examine other information that may be inferred from image tiles. For example, we show classification results for tiles, such as Luminal A versus Luminal B, with an F1 score larger than 0.85. We also show that Ki67 levels can be inferred from H&E tiles, as shown before on other cohorts, and that inference is still possible when working with lower dimensional latent representations. The two main contributions of this paper are as follows. First, demonstrating that the use of image tiles can be informative, both at the global classification level, and, more importantly, to support the assessment of heterogeneity. Second, reasonably accurate inference can be performed with lower dimensional latent representations of the H&E images.

available at https://www.kaggle.com/datasets/chloebnms/h-and-e-images-breast-cancer-cohort.

**Funding:** The RESCUER project has received funding from the European Union's Horizon 2020 Research and Innovation Programme under Grant agreement No. 847912. A.P. received funding from Fundació La Marató TV3 201935-30, Fundación CRIS contra el cáncer PR_EX_2021-14, Agència de Gestó d'Ajuts Universitaris i de Recerca 2021 SGR 01156, Fundación Fero BECA ONCOXXI21, Instituto de Salud Carlos III PI22/01017, Asociación Cáncer de Mama Metastásico IV Premios M. Chiara Giorgetti, Breast Cancer Research Foundation BCRF-22-198 and BCRF-23-198, and RESCUER, funded by European Union's Horizon 2020 Research and Innovation Programme under Grant Agreement No. 847912. F.B-M. received funding from Fundación científica AECC Ayudas Investigador AECC 2021 (INVES21943BRAS). The funders had no role in study design, data collection and analysis, decision to publish, or preparation of the manuscript.

**Competing interests:** I have read the journal's policy and the authors of this manuscript have the following competing interests: A.P. reports advisory and consulting fees from Roche, Pfizer, Novartis, Amgen, BMS, Puma, Oncolytics Biotech, MSD, Guardan Health, Peptomyc and Lilly, lecture fees from Roche, Pfizer, Novartis, Amgen, BMS, Nanostring Technologies and Daiichi Sankyo, institutional financial interests from Boehringer, Novartis, Roche, Nanostring, Sysmex Europa GmbH, Medica Scientia inno. Research, SL, Celgene, Astellas and Pfizer; stockholder and consultant of Reveal Genomics, SL; patents filed PCT/EP2016/080056, PCT/EP2022/086493, PCT/EP2023/060810, EP23382703 and EP23383369. Z.Y. is consulting at Verily Inc. F.B-M. has patents filed: PCT/EP2022/086493, PCT/EP2023/060810, EP23382703 and EP23383369. The rest of the authors has no competing interests.

## Introduction

Pathology and the examination of results from cell labeling and staining assays play a central role in diagnosis, and classification of disease, especially cancer [1]. Traditional pathology has been applied in clinical practice since the mid 1870's [2,3]. Technological advances and the increased focus on precision medicine have recently paved the way for the development of automatic computer vision approaches to assist and complement traditional pathology methods. More specifically, solutions and methods are being developed, which allow us to explore and extract information beyond human visual perception and in faster rates [4,5]. In oncology, the application of machine learning and computer vision methods enables researchers and practitioners to more accurately and efficiently decipher complex pathophysiologies and to accelerate the discovery of novel biomarkers, including for personal treatment protocols, addressing selection between numerous therapeutic routes, based on patient molecular profiles [6–10]. In Baxi et al. [11], the authors provide an excellent overview of recent progress and of the limitations involved in using machine learning approaches in biomarker discovery and in the actual clinical practice of personalized treatment. In particular - the authors argue that "the traditional role of pathologists in delivering accurate diagnoses or assessing biomarkers for companion diagnostics may be enhanced in precision, reproducibility, and scale by AI-powered analysis tools."

One important aspect of inferring therapeutic routes from patient molecular profiles is the tumor microenvironment and its heterogeneity, both in terms of cancer clonal populations and in terms of other cell types. Heterogeneity should be considered at the macro level tumor context (different parts of the tumor) [12,13] as well as at the microenvironment level [14–17]. In-situ molecular profiling approaches [18–21] are the most precise high resolution measurement tools for investigating the molecular composition from tumor biopsies, in a spatial context. H&E whole-slide images (WSIs) hold the potential to actually contain clinical information about the samples that is not covered by the standard interpretation procedures and protocols. H&E staining is also the most standard and cost effective procedure which is applied to almost all biopsied patient material. Recent work has demonstrated several example traits that can be partially associated to H&E images, and therefore probably to cell morphology, going far beyond the traditional scope of human interpretation. In Coudray et al. [22], the authors used deep learning on WSIs to classify samples into lung cancer subtypes. They also found that six commonly mutated genes in lung adenocarcinoma can be predicted from pathology images. In Levy-Jurgenson et al. [10], the authors trained deep learning models to spatially resolve bulk mRNA and miRNA expression levels within WSIs. Using this, the authors developed a method to quantify tumor heterogeneity. In Bergenstråhle et al. [23], the authors used deep learning techniques to characterize the transcriptome of micrometer-scale anatomical features and to predict spatial gene expression from histology images alone. In K de Haan et al. [24], the authors used supervised learning to transform from common H&E images to special stains. They showed that this method improves the diagnosis of several non-neoplastic kidney conditions, saving time and cost. A review of progress in this direction can be found in Shmatko et al. [25]. In [26], the authors demonstrate the use of autoencoders in analyzing spatial transcriptomics to address heterogeneity. In [27], autoencoders are used on IHC results as input. Approaching heterogeneity by using H&E images presents an even greater challenge.

Progress in using machine learning in digital pathology, specifically in H&E, only provides initial indications and will certainly not replace accurate measurement through specifically designed assays, such as IHC (Immune HistoChemistry). Nonetheless, machine learning

analysis of H&E WSIs can greatly enhance clinical practice. For example, it can serve to prioritize next steps, including the selection of the best IHC panel to use [28].

In this work, we show that by using a generic latent representation learned for H&E tiles selected biological characteristics are partially retained. We investigate this question using a richly annotated breast cancer cohort by employing simple autoencoders [29–32] with varying latent dimensions. The data for the cohort includes 106 postmenopausal women with stage I–IIIA hormone receptor-positive, HER2-negative and luminal B (by PAM50) breast cancer from the CORALLEEN trial. The patients were randomly assigned to one of two groups. The first one received the treatment investigated by Prat et al. [33]: ribociclib and daily letrozole. The second group received chemotherapy, specifically using doxorubicin, cyclophosphamide and paclitaxel. Samples were collected at three different timepoints; baseline (day 0), day 15, and surgery. The goal of the study of Prat et al. [33] was to assess the biological and clinical activity of the combination of ribociclib plus letrozole and how it compared to chemotherapy. The dataset contains biological details such as PAM50 subtype as infered through a gene expression profiling step or ROR (Risk of Recurrence) scores, gene expression markers such as MYC or FOXA1, along with the treatment regimens and corresponding patient response at each period.

The rest of the paper is constructed as follows. In the first section, we explain our methods and the different algorithmic steps. Then, we present our results, in particular, we show how image tiles can be classified as Ki67 high and low and how this can be done from a lower dimensional latent representation of the tiles. Finally, in the last section, we discuss open issues and future work.

## Materials and methods

### Data

We use data from the SOLTI CORALLEEN dataset [33], where the authors report measurements in breast cancer biopsies at three different timepoints: before treatment, after 3 weeks of treatment and at surgery. We use samples for 106 patients. Images are partitioned into tiles of shape 128x128x3. We keep the position of each tile in the filename.

The original study [33] was approved by the Ethics Committee at Hospital Clinic of Barcelona (HCB.2020.1233) and all methods were carried out in accordance with relevant guidelines and regulations. This study involves the use of H&E staining images from tissue samples of patients that were included in the CORALLEEN trial. These samples are stored in the biorepository of the Translational genomics and targeted therapies in solid tumors at IDIBAPS as long as patients sign the specific informed consent of the collection. Adequate written consent was obtained for the original study [33].

This article is a retrospective study on data previously studied. The data was first accessed on November 18th, 2022, and authors didn't have access to information that could identify individual participants during or after data collection, as samples were anonymized.

### Direct classification from tile to trait

We started by asking whether we could use deep learning to classify the tiles into different cancer-related traits. In particular, we looked at the traits: Ki67, TILS, MYC, FOXA1, HIF1A, ROR Subtype, ROR Subtype and Proliferation, PR status, PAM50 subtype. These traits are either continuous or binary.

For continuous traits, we defined the values below the 20th percentile to be 0 and the values above the 80th percentile to be 1. We then defined a binary labeling. We also defined

an Out-Of-Distribution (ood) test on the images for which the labels were between the 20th and 30th quantile, classified as 0, and between the 70th and 80th quantile, classified as 1. When splitting the images into train, validation and test set, we ensured that all tiles from the same WSI are assigned to the same set. Note that the split need not be per patient (all tiles from three WSIs assigned to the same set) since each of the three samples for a given patient was taken at a different timepoint. This is different to previous work [10,22] where it was possible for a patient to have two adjacent slides.

To develop the models, we first normalized the data and tried pretrained networks for transfer learning: Resnet50, Densenet121 and InceptionV3. All the networks were trained on the ImageNet dataset. We used the Binary Cross Entropy with Logits loss function and Adam optimizer with a learning rate of 0.001 for each network and a batch size of 8. After splitting into train, validation and test sets, we ran the model on the validation set for 20 epochs and selected the model that gives the best accuracy on this validation set. Each set is downsampled by randomly taking the same number of tiles from each label group as in the smallest set. We then ran the selected model on the test set and the ood set to see how it performed on unseen data.

## Classification from encoded tile to trait

In the second part, we tested whether encoding the tiles to a low-dimensional latent space and running the models on the encoded tiles would still yield good results.

For this task, we used only the data that is below the 20th quantile and above the 80th quantile. We first trained an autoencoder to recreate the original image using the following latent dimensions: {16384, 4096, 512, 64, 2}. We ran the model for 40 epochs and chose the model that gives us the smallest loss on the reconstruction of the images from the validation set. For this part, we use the MSE loss as a criterion, Adam optimizer with a learning rate of 0.001 and batch size of 16. We then used the same pre-trained networks and hyperparameters as in the direct classification task, but used as input the encoded tiles. We performed this analysis only for Ki67.

## Inertia using smaller latent space

In this section, we investigated the inertia of the encoded tiles with respect to the actual trait binning and compared it to random labeling/binning. We used the following formula to evaluate the inertia of any given binning, in any latent space:

$$J(C) = \frac{1}{m} \sum_{i=1}^{m} ||\vec{x}_i - \mu(C(\vec{x}_i))||$$

Where:

- $\vec{x}_i$: the latent representation for tile sample $i$
- $C$: a function that provides the class/bin/label for each tile
- $m$: total number of instances
- $\mu(C(\vec{x}_i))$ : center of class/bin/label to which sample i belongs, also a vector in the latent space
- The norm is Euclidean 2-norm

To get a notion of whether the encoding preserves label categories, we:

- Shuffled the labels of the original images, before tiling (to keep the proportions)
- Used the above formula to get the evaluation for the inertia of the newly created random labels. (The $\mu$s are recomputed with the new labels each time).

We performed this 100 times to get a distribution and compared it with the actual value of $J(C)$.

## Results

We used the SOLTI CORALLEEN dataset [33], pertaining on 283 breast cancer biopsies of 106 patients at three different timepoints. These biopsy images are segmented into tiles. A tile, in this context, is a small part of an image. Specifically - the H&E images, which are high-resolution images, are divided into tiles which are 128 by 128. The distribution of the number of tiles is presented in S1 Fig. More details about the tiling process can be found in S2 Fig.

For each biopsy image, we obtained measurements for:

- Ki67 – a protein marker associated with cellular proliferation, indicating how rapidly cancer cells are dividing. It is determined at the protein level in FFPE tumor tissue sections by immunohistochemistry.
- TILs (Tumor-Infiltrating Lymphocytes) – represent the immune cells found within the tumor and constitute an important indicator of the immune response to cancer. It is determined in FFPE tumor tissue sections by H&E staining.
- PR (Progesterone Receptor). It indicates the presence or absence of progesterone receptors on cancer cells, helping to determine hormone sensitivity and treatment options. We consider the PR value, which is continuous, and the PR status, which is either negative or positive. The PR value is determined at the protein level in FFPE tumor tissue sections by immunohistochemistry. The PR status is determined at the gene expression level in RNA extracted from FFPE tumor tissue using the nCounter platform.
- MYC – an oncogene that regulates gene expression and promotes cell growth and proliferation.
- FOXA1 – a transcription factor that is involved in hormone response and cellular differentiation.
- HIF1A – a regulatory protein that helps cells adapt to low-oxygen conditions, and is often upregulated in tumors.
- PAM50 subtype. A gene-expression-based procedure that is used to classify breast cancer into one of five subtypes: Luminal A, Luminal B, HER2-enriched, Basal-like and Normal-like. In this study we are only considering Luminal A and Luminal B.
- ROR Subtype. This represents the Risk of Recurrence based only on the PAM50 subptype. We denote it by ROR S.
- ROR Subtype and Proliferation. It represents the Risk of Recurrence score based on both the PAM50 subtype and the proliferation score. We denote it by ROR S & P.

MYC, FOXA1, HIF1A, PAM50 and both ROR are determined at the gene expression level in RNA extracted from FFPE tumor tissue using the nCounter platform.

Our goal is to train a deep learning classification network and predict those traits, using tiles created from the biopsy images. These traits, inherited from the image labels, are now our labels for the classification task, as in [10,22]. We binarize the values for these traits before proceeding as described in the Methods section.

To train our algorithm, we split the data into train, validation and test sets and map the tiles of each image to the assigned label, which is inherited from the slide's bulk label. For details on how we perform this split see the Methods section.

For each of the traits that have continuous values, we also define an out-of-distribution (ood) test set which consists of: (1) all tiles with bulk-labels between the 20th and 30th percentiles, which we binarize to a label of 0, and (2) all tiles with bulk-labels between the 70th and the 80th percentile, which we binarize to a label of 1.

To create our deep learning classification network, we used the pretrained networks Resnet50, Densenet121 and InceptionV3, which were all pretrained on the ImageNet dataset. These networks are widely used for image classification tasks. The final weights chosen for each network was the one that performed best on its validation set.

Note that the number of tiles obtained from each slide may be different due to their different sizes. To ensure balanced datasets, we downsample the data by randomly choosing the same number of samples from each label group as in the smallest set. We use this downsampled set to measure the algorithm's accuracy.

The remainder of the results section proceeds as follows: in the first part, we use classic deep learning classification architectures to classify each tile and use the predictions to produce heterogeneity maps. In the second part, we investigate whether using autoencoders retains the classification information. In the third part, we further study the latent representations.

## Direct classification from image tiles

In this section, we present the results of using a classification network directly from tile to label. For each trait, we select the best model as measured on the validation set, and calculate the scores of this model on the test and ood sets. In Table 1 we present the scores obtained on the validation set, and in Table 2 the scores we obtained on the test and ood sets.

One aim of this study is to assess how H&E can be used to determine molecular heterogeneity. To observe the heterogeneity, we take the test-set tiles and create a reconstruction of the full images using our neural network tile predictions.

We present in Fig 1 examples of heterogeneity map reconstructions and their images from which they were derived. For example, consider the trait PAM50, which, in our study has two labels, Luminal A and Luminal B. We have 28 images in the test set, where 14 are Luminal A and 14 are Luminal B. For Luminal A, 9 out of 14 images have more than 70% accuracy

**Table 1**. **Direct classification scores for different traits, on the validation set.** Set size represents the number of tiles. Numbers higher than 0.75 are in bold.

| Trait | Network | Validation set size | Accuracy | AUC-ROC Score | F1 Score |
|---|---|---|---|---|---|
| Ki67 | Densenet121 | 7,926 | **0.7875** | **0.8169** | **0.8110** |
| TILS | InceptionV3 | 19,786 | 0.6311 | 0.6679 | 0.5942 |
| PR Value | Densenet121 | 2,222 | 0.6539 | 0.6839 | 0.5800 |
| MYC | InceptionV3 | 7,818 | **0.7855** | **0.8453** | **0.7849** |
| FOXA1 | Resnet50 | 19,180 | 0.7010 | 0.7228 | 0.6580 |
| HIF1A | InceptionV3 | 11,822 | 0.6713 | 0.7170 | 0.6748 |
| ROR S | InceptionV3 | 8,404 | **0.8353** | **0.8787** | **0.8471** |
| ROR S & P | InceptionV3 | 7,594 | **0.8770** | **0.9278** | **0.8799** |
| PR status | Densenet121 | 2,394 | 0.5459 | 0.5103 | 0.5446 |
| PAM50 | InceptionV3 | 25,262 | 0.7468 | **0.8115** | **0.7653** |

**Table 2. Direct classification scores for different traits, on test and ood set.** Note that performance on the ood set is close to random for most traits. Numbers higher than 0.75 are in bold.

| Trait | Test set | | | | Ood set | | | |
|---|---|---|---|---|---|---|---|---|
| | set size | Accuracy | AUC-ROC | F1 | set size | Accuracy | AUC-ROC | F1 |
| Ki67 | 2,934 | **0.8224** | **0.9012** | **0.8243** | 27,586 | 0.5668 | 0.6027 | 0.5455 |
| TILS | 17,146 | 0.6640 | 0.7265 | 0.6365 | 178,104 | 0.5530 | 0.5680 | 0.4852 |
| PR value | 788 | 0.6396 | 0.7092 | 0.5919 | 6,036 | 0.4468 | 0.4242 | 0.3720 |
| MYC | 9,714 | 0.5566 | 0.6335 | 0.5645 | 37,286 | 0.6059 | 0.6652 | 0.6052 |
| FOXA1 | 5,232 | 0.6460 | 0.6872 | 0.6333 | 28,894 | 0.7120 | **0.7691** | 0.7097 |
| HIF1A | 10,318 | 0.5927 | 0.6304 | 0.6160 | 34,266 | 0.5889 | 0.6168 | 0.6137 |
| ROR S | 4,916 | **0.8942** | **0.9447** | **0.8943** | 39,134 | 0.6242 | **0.7521** | 0.4422 |
| ROR S & P | 3,510 | 0.7091 | 0.7000 | **0.7521** | 37,142 | 0.5696 | 0.5943 | 0.4734 |
| PR status | 2,168 | 0.4958 | 0.4511 | 0.4499 | | | | |
| PAM50 | 10,636 | **0.8491** | **0.9185** | **0.8545** | | | | |

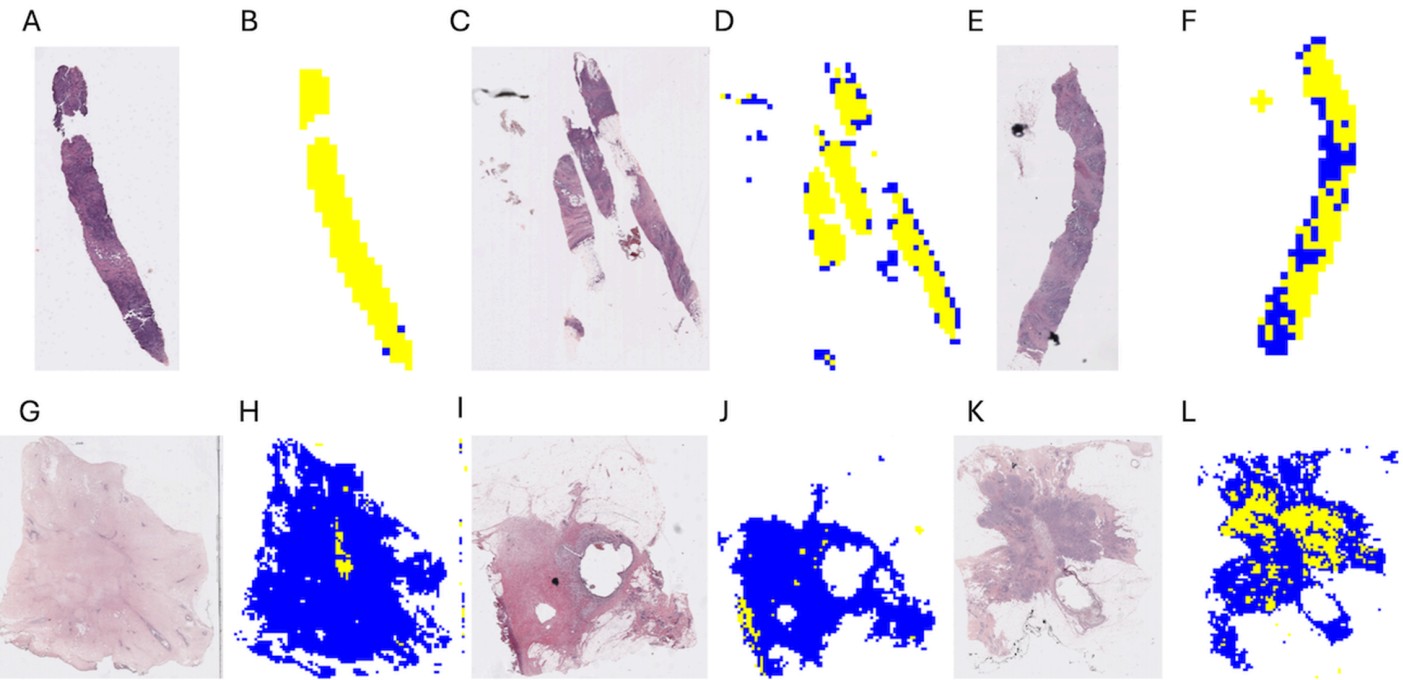

**Fig 1. Luminal A vs Luminal B heterogeneity map reconstructions, using predicted tile labels, and their original images.** (A) Original image. (B) Inferred heterogeneity map. Luminal B sample, with 0.9874 accuracy. (C) Original image. (D) Inferred heterogeneity map. Luminal B sample, with 0.7553 accuracy. (E) Original image. (F) Inferred heterogeneity map. Luminal B sample, with 0.6029 accuracy. (G) Original image. (H) Inferred heterogeneity map. Luminal A sample, with 0.9795 accuracy. (I) Original image. (J) Inferred heterogeneity map. Luminal A sample, with 0.9615 accuracy. (K) Original image. (L) Inferred heterogeneity map. Luminal A sample, with 0.7376 accuracy.

on their tile label. For Luminal B, 11 out of 14 images have more than 75% accuracy on their label, and all of them have more than 60%.

Yellow is Luminal B prediction and blue is Luminal A prediction. We observe that while an accurate label is inferred for most tiles, some tiles deviate from that behaviour, possibly representing actual heterogeneity. It is important to note that the use of tiles yields information that is beyond the global sample label. In particular, it allows for the assessment of heterogeneity, e.g. by using the methods of Levy-Jurgenson [17].

## Autoencoders

Following this section, we investigate the effect of the latent space dimension on classification at the tile level. To do so, we use autoencoders with different latent space dimensions. In this section, we describe the autoencoders. In general, the advantage of using lower dimensional representation may be the denoising effect that it facilitates. Also, the intrinsic dimension of the data is not clear, therefore, one may need to explore different latent space dimensions.

The original tile dimension is 128*128*3 = 49,152, representing 128*128 pixels, with RGB for each one. In the investigation of latent representations, we focus on the trait Ki67 [10] as a case study. Ki67 is one of the IHC procedures that are part of standard diagnosis and treatment in most cancer types [34–36]. We have 48,746 elements (tiles) in the training set, and 7,926 elements in the validation set.

The first autoencoder that we present is a simple autoencoder that consists of 3 conv2D layers. The latent space dimension is of size 64*16*16=16,384 We present in Fig 2a example images and corresponding reconstructions. In the second autoencoder, we have 5 conv2D layers, leading to a latent space dimension of size 256*4*4=4096. We present these results in Fig 2b. In the third autoencoder, we have 6 conv2D layers, for a latent space dimension of size 128*2*2=512. We present the reconstruction in Fig 2c. In the fourth autoencoder, we have 7 conv2D layers, giving us a latent space of size 64*1*1=64. We present the reconstruction in Fig 2d. Finally, in our last autoencoder, we also have 7 conv2D layers, where our latent space is 2*1*1=2. We present the reconstruction in Fig 2e. As we can see, the reconstruction performances degrades as the latent space dimension is reduced. We present in Fig 3 a boxplot of the MSE loss of the validation set across all the dimensions studied.

## Classification from encoded tile to trait

Once we have our autoencoder, we want to encode our data and run a binary classification on these encoded images. This can give us a fair idea whether the information is still contained in the encoded image. We test this approach for Ki67. We have 48,746 tiles in the training set, 7,926 tiles in the validation set, 2,934 tiles in the test set and 27,586 tiles in the ood set.

We test the following dimensions: 16,384, 4,096, 512, 64 and 2. We present in Table 3 the results for the validation set, Fig 4 the confusion matrices for the test set and ood set.

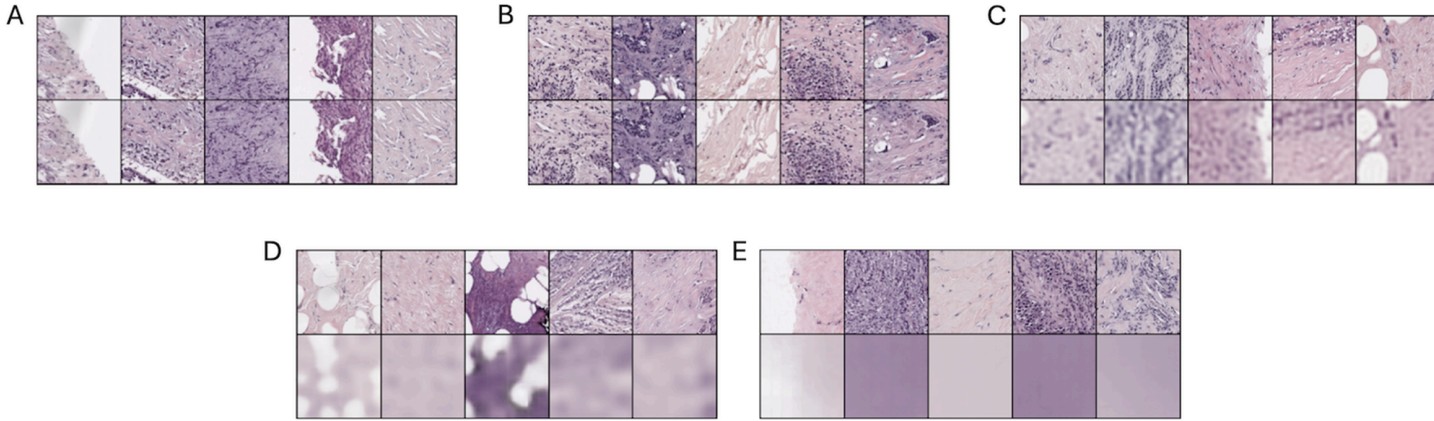

**Fig 2. Original images and corresponding reconstruction based on an autoencoder for latent space dimensions of 16,384, 4,096, 512, 64, and 2.**

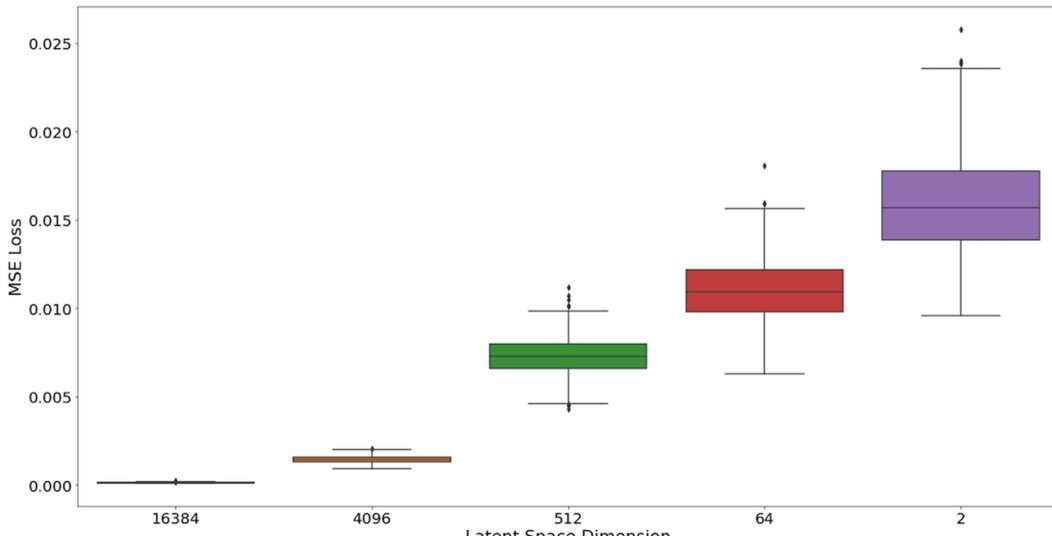

**Fig 3. Boxplot representing the MSE loss in the validation set according to the latent space dimension of the autoencoder.**

**Table 3. Validation scores for the different latent spaces, label is Ki67.**

| Latent Space Dimension | Network | Validation Accuracy | AUC-ROC Score | F1 Score |
|---|---|---|---|---|
| No encoder | Densenet121 | **0.7875** | **0.8169** | **0.8110** |
| 16,384 | Resnet50 | **0.7918** | **0.8506** | **0.8113** |
| 4,096 | Resnet50 | **0.7792** | **0.8285** | **0.8063** |
| 512 | Resnet50 | 0.7257 | **0.7548** | 0.7295 |
| 64 | Densenet121 | 0.7215 | 0.7378 | 0.7157 |
| 2 | Densenet121 | 0.6578 | 0.6884 | 0.6471 |

As expected, as we decrease the dimension of the latent representation, the AUC also decreases. However, we also see that much of the information is retained at lower dimensions. This demonstrates the potential for more efficient data analysis using lower dimensional latent representations.

In Fig 5, we present a summary of the validation values and the test values across all the different latent space dimension. We also depict a 95% confidence interval for the observed accuracies. Recall that we have a sample size of 7,926 tiles in the validation set and a sample size of 2,934 tiles in the test set. We can see that the accuracy decreases with the latent dimension.

We now create a reconstruction of the full test image or, alternatively, the inferred IHC map, using prediction of each tile using our autoencoder and neural network. We present in Fig 6, the original image, the reconstructed image with no encoder and with encoders of size 16,384, 4,096, 512, 64 and 2, for three different examples.

By looking at the Ki67 inferred maps, we can see that we obtain different results and accuracy at each latent space dimension. Even if the accuracy is maintained in lower dimensions, most of the information related to the heterogeneity is lost.

In future work, we can compare the different reconstruction to a heterogeneity calculation of Ki67, computed e.g. from IHC and see up to which dimension the heterogeneity is still sustained. It is important to note that the accuracy numbers don't really reflect tile level

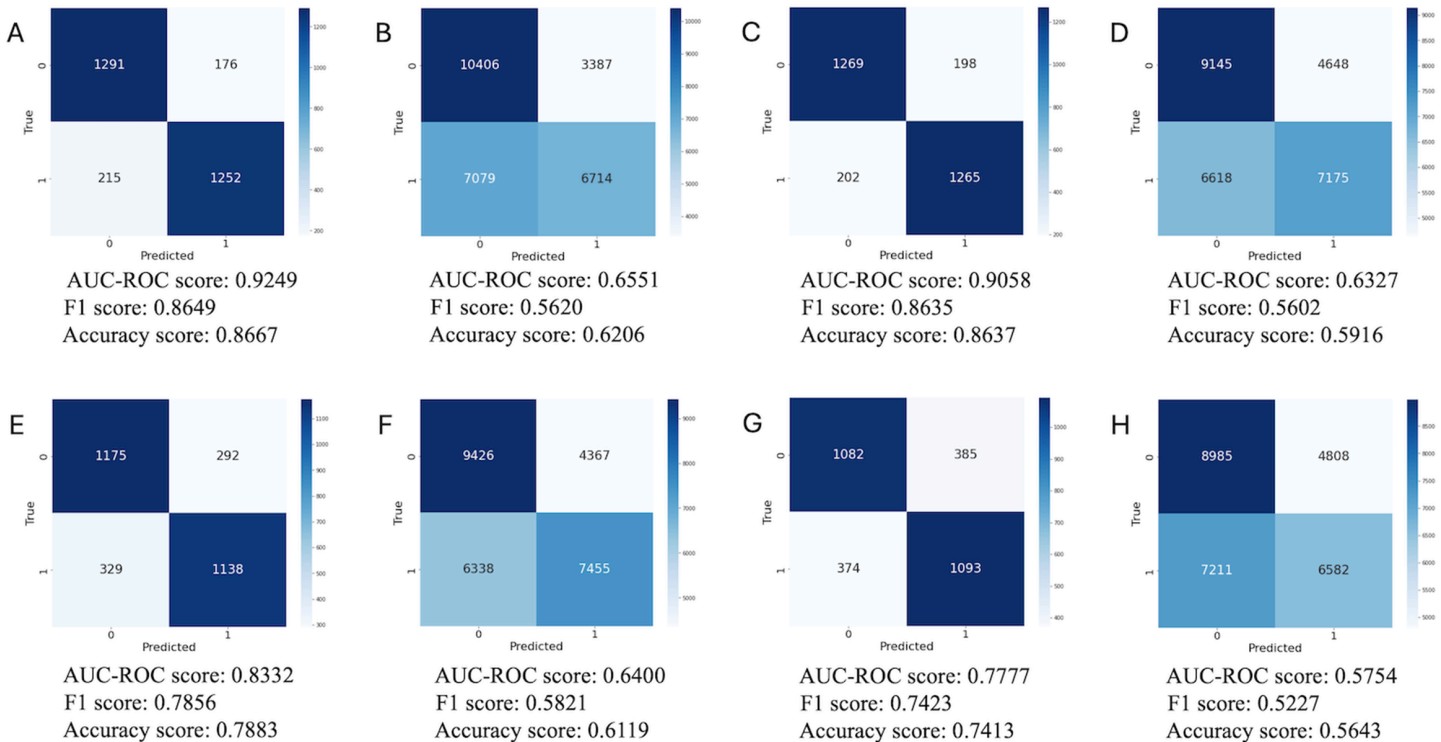

**Fig 4. Confusion matrices and corresponding performance scores for selected latent space dimensions on test and ood sets. Label is Ki67.** (A) Confusion matrix for latent space 16,384, on test set. (B) Confusion matrix for latent space 16,384, on ood set. (C) Confusion matrix for latent space 4,096, on test set. (D) Confusion matrix for latent space 4,096, on ood set. (E) Confusion matrix for latent space 512, on test set. (F) Confusion matrix for latent space 512, on ood set. (G) Confusion matrix for latent space 2, on test set. (H) Confusion matrix for latent space 2, on ood set.

accuracies as the labels are given for the entire image/sample and therefore not all tiles are correctly Ki67 labeled in the ground truth.

We also compare the heterogeneity mappings to images annotated by pathologists. We present in Fig 7 some examples of comparison between our results and the annotated images. We can see that the tiles classified as high Ki67 are most of the time also highlighted by the pathologists as tumor regions.

## Inertia using smaller latent space

The inertia is a function that measures the cohesiveness of any given input partition of a set of elements into subsets. It is often used as an optimization target for clustering algorithms, such as k-means. See Methods for the full definition.

We calculate the inertia of binning tiles according to actual image-level Ki67 (or other traits). We aim to show that clinical labels obtain better inertia values, in any latent representation, as compared to random labeling. To this end, we investigate all latent dimensions: 16,384, 4,096, 512, 64, 2. For each latent space dimension, we take each label as presented in the first part of the paper, and split the continuous labels in three categories: 0-33th percentile, 33-66th percentile and 66-100th percentile. We keep the binary labels as they are. This defines our label for each trait and tile. In each step, we select 5 images belonging to each label, and 100 tiles from each. Each tile is encoded at the dimension we are investigating. We calculate the inertia of the tiles by using their labels as the binning criterion, and compare it to

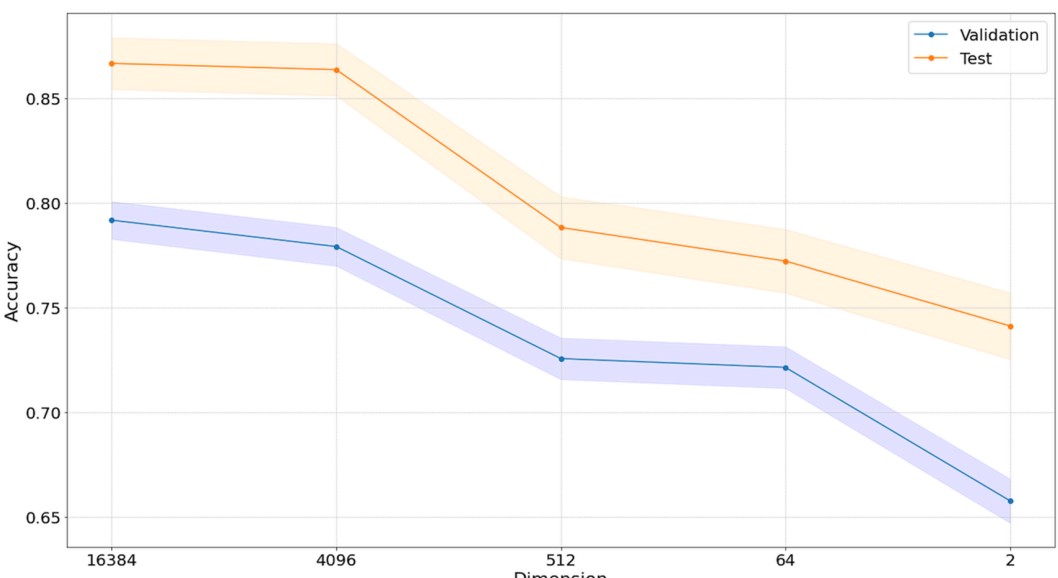

**Fig 5. Validation accuracy and test accuracy and their 95% confidence intervals across the different latent space dimensions. Note that the sets are balanced.**

random binning, keeping the proportions. We do this operation 100 times to obtain a distribution. We present in Fig 8 the distributions of the random labeling (in blue) for latent space 16,384 and 64, compared to the true Ki67 labeling of the data (red line).

From this, we get the percentage of elements from the random distribution which have a higher distance inertia obtained for binning using the actual labels. We present these results in Fig 9. We find that the proportions are kept, independent of the size of the latent space dimension.

## Discussion

In this work, we presented an analysis of H&E images using deep learning classification and autoencoders. We reported results and methods in three different, but related, aspects. In the first part, we showed that some traits can be inferred from direct classification from tiles. In the second part, we showed that when encoding the tiles to a low-dimensional latent space, we partially retain the information that was contained in the original tile. Finally, in the last part, we compared the inertia of the original labeling to random labeling using different latent space dimensions. A long term future vision of developing latent representation models for WSIs is to use these to enable a more effective assessment of tumor heterogeneity – an important aspect of resistance to therapy. In this work, we present results, as indicated above, that contribute to substantiating the validity of this vision.

One surprising aspect of our result is the fact that the sample level information was mostly retained when reducing the latent space dimension even as far as dimension = 2, as we can see in Figs 4g and 9. Even though classification performance is not as good as in higher dimensions, we still see signals even in dimension = 2. This fact can possibly be attributed to tumor cell density which is associated with both the color density and the levels of some of the traits, such as Ki67. We also see that for some traits, the lower dimensional representation sustains the information contained in the tiles, while for others, such as the PR status for example, neither the original H&E images nor the encoded representations contain the information.

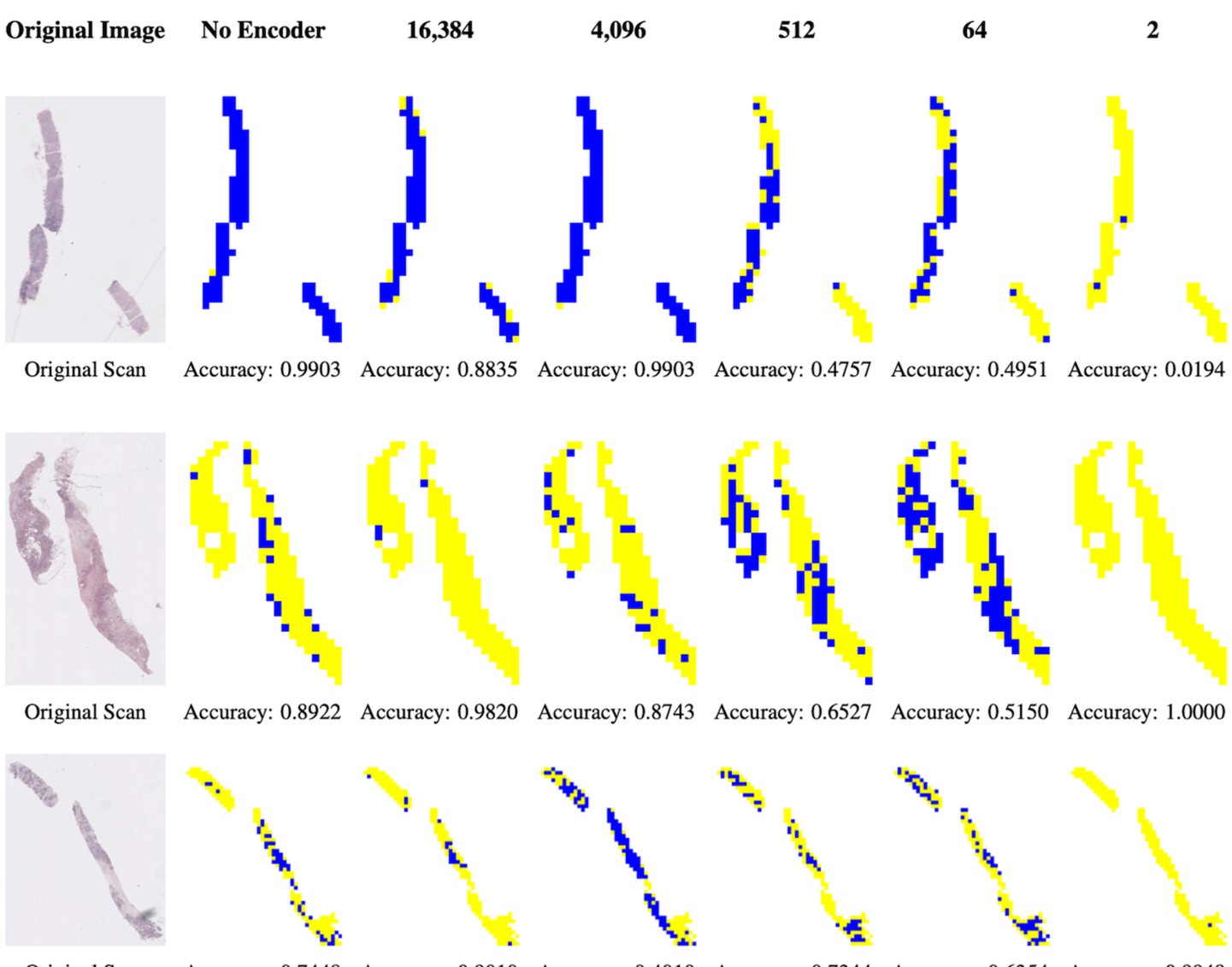

**Fig 6. Original images and their inferred Ki67 maps across the different latent space dimensions.** Yellow represents low Ki67, and blue represents high Ki67. It is important to note that the use of tiles yields information that is beyond the global sample label. In particular, it allows for the assessment of heterogeneity, e.g. by using the methods of Levy-Jurgenson [17]. Also note that whole image level classification results might change with the latent dimension. Determining an adequate operational dimension remains open (see Discussion).

This probably indicates that PR does not affect cell morphology. We also note that working with different latent dimensions can lead to different conclusions in terms of sample level classification and heterogeneity, see Fig 6. As, in this work, tile level labels are inherited from the sample, the accuracy reported in Fig 6 does not really reflect comparison to the ground truth. In some dimensions we would infer different heterogeneity scores in different dimensions, possibly also affecting the nominal accuracy. One limitation of our approach is therefore that the choice of dimension is not currently guided by any rigorous principles. Better understanding of thresholds and effects is a direction for further research. In future work, we also intend to focus on the heterogeneity itself. Using the encoded tiles can help us define

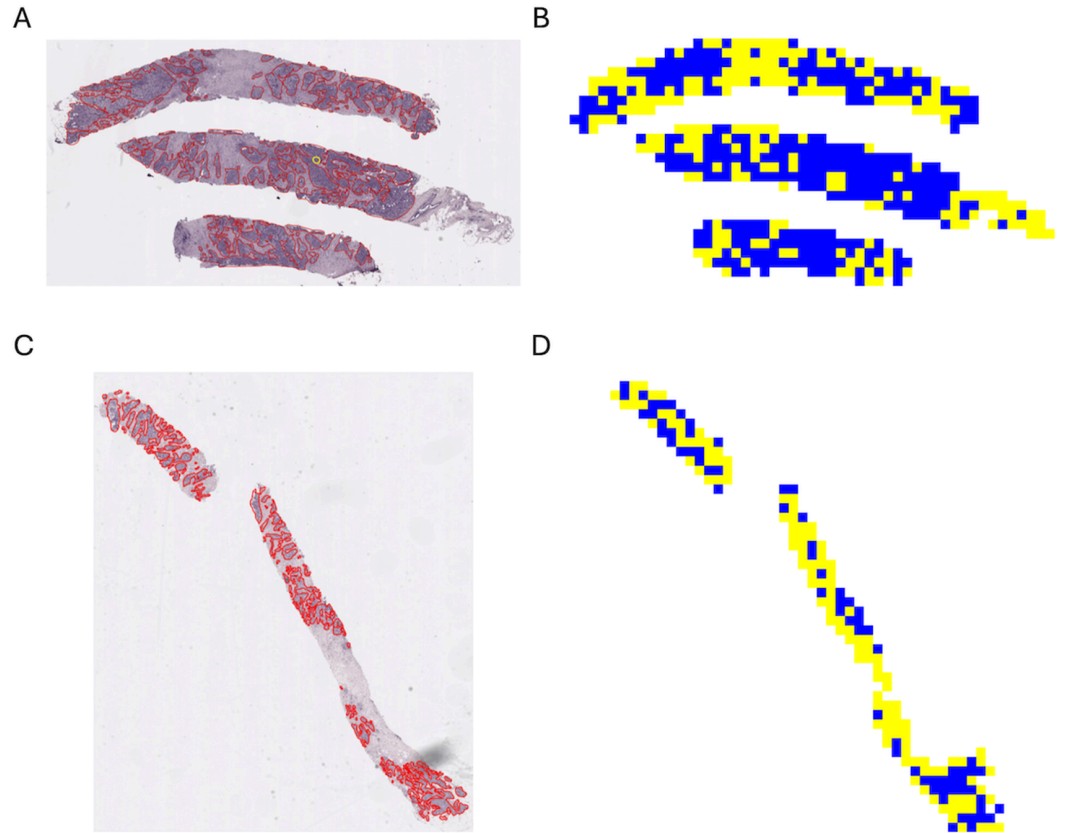

**Fig 7. Comparison of original annotated images vs the reconstruction of their encoded tiles.**

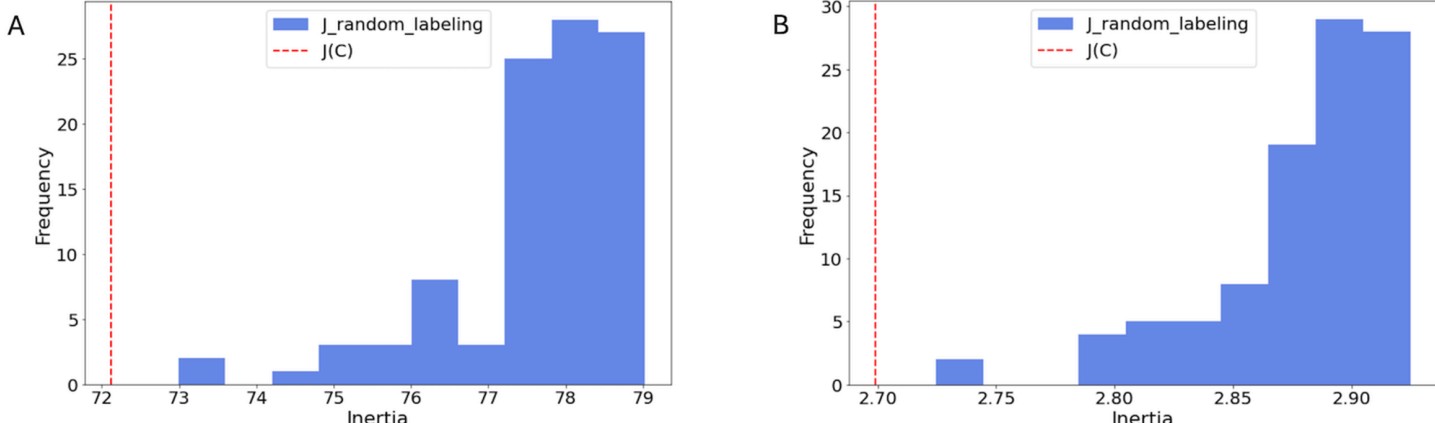

**Fig 8. Inertia of the latent representation. Dimensions = 16,384 and 64.** (A) Comparison of the inertia computed for binning latent representation vectors according to Ki67 (red line) vs the inertia values of 100 random labeling instances. Latent space dimension = 16,384 (blue histogram). (B) Comparison of the inertia computed for binning latent representation vectors according to Ki67 (red line) vs the inertia values of 100 random labeling instances. Latent space dimension = 64 (blue histogram).

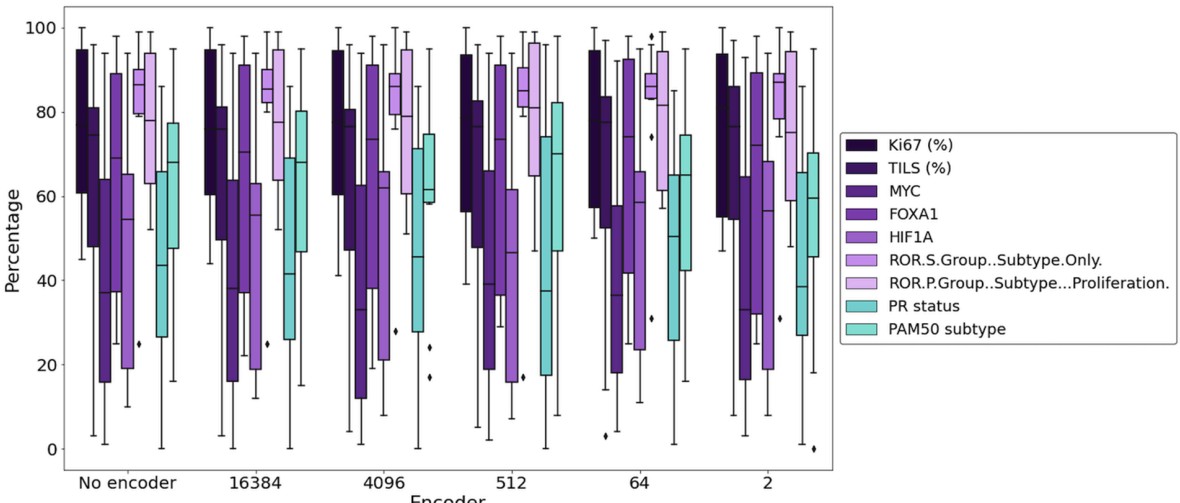

**Fig 9. Percentage of instances for which the random inertia is above the inertia computed for the indicated traits and dimensions. Purple - trinary labels. Green - binary labels.**

heterogeneity of the full biopsy image and without the need to pre-determine the traits over which heterogeneity is measured. This is the purpose of the inertia data as presented in the Results section. These data show that for significant clinical characteristics of tiles, we can expect proximity in the latent space for tiles with similar values. The data shows that this is true for the known clinical parameter Ki67. As we use shuffling at the full image level (all labels in the same image stay the same after shuffling), this observed proximity transcends that which we would expect due to image level properties being similar for neighboring tiles. The hypothesis is that this would also be true for unspecified clinical characteristics and can therefore support the determination of heterogeneity.

To further explore the effect of the latent space on the separation of different classes, we performed two dimensional t-SNE on the latent representation of the data. The results are described in S3 Fig.

In summary, the two main contributions of this paper are as follows. First, we demonstrate that the use of image tiles can be informative, both at the global classification level, and, more importantly, if tile level classification of certain traits is sufficiently good, to support the assessment of heterogeneity based on these traits. Second, we show that reasonably accurate inference can be performed with lower dimensional latent representations of the H&E images.

## Supporting information

**S1 Fig. Distribution of the number of tiles per image.** We analyzed a total of 283 images from 106 patients. The number of tiles varies due to the characteristics of the H&E staining and the microscopy. Namely - white space in the H&E images is ignored and not included in the tiles.
(TIF)

**S2 Fig. Visual representation of the tiling process and tile selection.** (A) Original biopsy image with the tissue outlined in black for better visualization. (B) Demonstration of the image tiling process (note: the actual tile size is significantly smaller in practice). (C) Tiles

highlighted in green are selected, while those in red are discarded (containing more than 50% background).
(TIF)

**S3 Fig. tSNE representation of different latent space dimension for Ki67, TILS and PR.** To create the tSNE representation, we define the label of Ki67, TILS and PR to be 0 if the label is less than the 20th quantile and 1 if the label is more than the 80th quantile. We select five images for each label, and from each image we select 100 tiles. We encode those 1,000 tiles to every dimension and show the tSNE representation of these encoded tiles. Color coded according to the 0/1 class. The figure depicts 10 different selections of the 10 images.
(ZIP)

## Acknowledgments

We thank the RESCUER consortium members for useful discussions and for the collaboration. We thank the Yakhini Research Group, especially Alon Oring, for important discussion and comments.

## Author contributions

**Conceptualization:** Chloe Benmussa, Alona Levy-Jurgenson, Zohar Yakhini.

**Data curation:** Esther Sanfeliu, Anabel Martinez-Romero, Blanca González-Farré, Tomás Pascual, Joaquín Gavilá, Fara Brasó-Maristany, Aleix Prat.

**Formal analysis:** Chloe Benmussa.

**Funding acquisition:** Aleix Prat.

**Investigation:** Chloe Benmussa, Ariel Shamir, Zohar Yakhini.

**Methodology:** Chloe Benmussa, Alona Levy-Jurgenson, Ariel Shamir, Zohar Yakhini.

**Project administration:** Fara Brasó-Maristany, Zohar Yakhini.

**Software:** Chloe Benmussa.

**Supervision:** Aleix Prat, Zohar Yakhini.

**Validation:** Alona Levy-Jurgenson, Ariel Shamir, Fara Brasó-Maristany, Zohar Yakhini.

**Writing – original draft:** Chloe Benmussa, Alona Levy-Jurgenson, Ariel Shamir, Zohar Yakhini.

**Writing – review & editing:** Chloe Benmussa, Alona Levy-Jurgenson, Ariel Shamir, Fara Brasó-Maristany, Zohar Yakhini.

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
