## [Decision Letter · Decision Letter 0]

11 Oct 2024

PONE-D-24-28161Latent representation of H&E images retains clinical information in a breast cancer cohortPLOS ONE

Dear Dr. Benmussa,

Thank you for submitting your manuscript to PLOS ONE. After careful consideration, we feel that it has merit but does not fully meet PLOS ONE’s publication criteria as it currently stands. Therefore, we invite you to submit a revised version of the manuscript that addresses the points raised during the review process.

Please accept our sincere apologies for the delay on this review. Please address the reviewers' comments below.

We look forward to receiving your revised manuscript.

Kind regards,

Amy McCart Reed

Academic Editor

PLOS ONE

**Comments from the Journal Office**

Please note that a second reviewer was invited to assess your manuscript, but was unable to submit their comments formally through Editorial Manager. However, they have provided valuable feedback on your manuscript, and their comments are available at the end of this email. Please provide your response to the comments of both reviewers in your rebuttal. 

“The RESCUER project has received funding from the European Union’s Horizon 2020 Research and Innovation Programme under Grant agreement No. 847912.

A.P. received funding from Fundació La Marató TV3 201935-30, Fundación CRIS contra el cáncer PR\_EX\_2021-14, Agència de Gestó d'Ajuts Universitaris i de Recerca 2021 SGR 01156, Fundación Fero BECA ONCOXXI21, Instituto de Salud Carlos III PI22/01017, Asociación Cáncer de Mama Metastásico IV Premios M. Chiara Giorgetti, Breast Cancer Research Foundation BCRF-22-198 and BCRF-23-198, and RESCUER, funded by European Union's Horizon 2020 Research and Innovation Programme under Grant Agreement No. 847912. F.B-M. received funding from Fundación científica AECC Ayudas Investigador AECC 2021 (INVES21943BRAS).”

“I have read the journal's policy and the authors of this manuscript have the following competing interests: A.P. reports advisory and consulting fees from Roche, Pfizer, Novartis, Amgen, BMS, Puma, Oncolytics Biotech, MSD, Guardan Health, Peptomyc and Lilly, lecture fees from Roche, Pfizer, Novartis, Amgen, BMS, Nanostring Technologies and Daiichi Sankyo, institutional financial interests from Boehringer, Novartis, Roche, Nanostring, Sysmex Europa GmbH, Medica Scientia inno. Research, SL, Celgene, Astellas and Pfizer; stockholder and consultant of Reveal Genomics, SL; patents filed PCT/EP2016/080056, PCT/EP2022/086493, PCT/EP2023/060810, EP23382703 and EP23383369. Z.Y. is consulting at Verily Inc. F.B-M. has patents filed: PCT/EP2022/086493, PCT/EP2023/060810, EP23382703 and EP23383369. The rest of the authors has no competing interests.”

5. In the online submission form, you indicated that [Code available upon request. For data access, please contact fbraso@clinic.cat. And for code access, contact chloebenmussa@gmail.com].

“We thank the RESCUER consortium members for useful discussions and for the collaboration. We thank the Yakhini Research Group, especially Alon Oring, for important discussion and comments. The RESCUER project has received funding from the European Union’s Horizon 2020 Research and Innovation Programme under Grant agreement No. 847912. A.P. received funding from Fundaci´o La Marat´o TV3 201935-30, Fundaci´on CRIS contra el c´ancer PR EX 2021-14, Ag`encia de Gest´o d’Ajuts Universitaris i de Recerca 2021 SGR 01156, Fundaci´on Fero BECA ONCOXXI21, Instituto de Salud Carlos III PI22/01017, Asociaci´on C´ancer de Mama Metast´asico IV Premios M. Chiara Giorgetti, Breast Cancer Research Foundation BCRF-22-198 and BCRF-23-198, and RESCUER, funded by European Union’s Horizon 2020 Research and Innovation Programme under Grant Agreement No. 847912. F.B-M. received funding from Fundaci´on cient´ıfica AECC Ayudas Investigador AECC 2021 (INVES21943BRAS).”

“The RESCUER project has received funding from the European Union’s Horizon 2020 Research and Innovation Programme under Grant agreement No. 847912.

A.P. received funding from Fundació La Marató TV3 201935-30, Fundación CRIS contra el cáncer PR\_EX\_2021-14, Agència de Gestó d'Ajuts Universitaris i de Recerca 2021 SGR 01156, Fundación Fero BECA ONCOXXI21, Instituto de Salud Carlos III PI22/01017, Asociación Cáncer de Mama Metastásico IV Premios M. Chiara Giorgetti, Breast Cancer Research Foundation BCRF-22-198 and BCRF-23-198, and RESCUER, funded by European Union's Horizon 2020 Research and Innovation Programme under Grant Agreement No. 847912. F.B-M. received funding from Fundación científica AECC Ayudas Investigador AECC 2021 (INVES21943BRAS).”

7. We notice that your supplementary figure 1 is included in the manuscript file. Please remove them and upload them with the file type 'Supporting Information'. Please ensure that each Supporting Information file has a legend listed in the manuscript after the references list.

Reviewers' comments:

Reviewer's Responses to Questions

**Comments to the Author**

1. Is the manuscript technically sound, and do the data support the conclusions?

Reviewer #1: Yes

2. Has the statistical analysis been performed appropriately and rigorously? 

Reviewer #1: I Don't Know

3. Have the authors made all data underlying the findings in their manuscript fully available?

Reviewer #1: Yes

4. Is the manuscript presented in an intelligible fashion and written in standard English?

Reviewer #1: Yes

5. Review Comments to the Author

Reviewer #1: The authors describe a computational approach to classify regions of Haematoxylin and eosin stained Whole Slide Images of breast cancer with corresponding biological/molecular labels and to use autoencoders to generate a latent (lower dimension) representation of the data, highlighting how this impacts retention of clinically relevant information. The study is relevant and the approach appears to be sound but I suggest that the term "clinical" in the title be replaced as the majority of the data appears to be directly tumour-related (pathological/molecular or biological).

In areas the language is somewhat unclear e.g. lines 273/274 making it difficult to understand. The paper would benefit from some language revision of the text in areas to improve readability and understanding.

6. PLOS authors have the option to publish the peer review history of their article (what does this mean?). If published, this will include your full peer review and any attached files.

Reviewer #1: No

**Addition Feedback from a Second Reviewer**

Manuscript Summary:

This paper investigates the use of convolutional autoencoders to reduce the compute requirements for performing deep learning classification on tumor histology images. First, this paper evaluates deep learning prediction capabilities for a series of biomarkers in the specific breast cancer dataset used. These results are reported, and the authors speculate on tumor heterogeneity using tile-level predictions of PAM50 subtype. Second, a series of convolutional autoencoders were trained to reduce the dimensionality of the tile images and evaluate whether the biomarker label Ki67 could be learned from this lower-dimensional data. Across a range of sizes of latent embeddings, Ki67 could still be predicted, although performance was diminished by reducing the latent embedding dimensions. Additionally, to investigate whether biomarker labels can be inferred from the latent embedding, the authors compare the inertia for a subset of tiles with biomarker labels to random labeling and find lower inertia for biomarker labeled tiles.

Significance:

Understanding the heterogeneity of cancer from a deep learning computation on readily-available H&E stained slides could be clinically significant, particularly for predicting which patients may have treatment-resistant cancers and adjusting their treatment strategy. Previous studies have explored the use of deep learning applied to tumor histology images to infer heterogeneity, including Levy-Jurgenson et al (2020, https://doi.org/10.1038/s41598-020-75708-z) and Zormpas-Petridis et al (2021, https://doi.org/10.3389/fonc.2020.586292) which used whole-slide labels to infer heterogeneity of tumors. 

Additionally, Shahamatdar et al (2024, https://doi.org/10.1111/his.15180) learned local tile predictions from whole-slide labels and compared these predictions to spatial labels from laser capture microdissection (LCM) to find that deep learning predictions were not consistent with LCM ground truth. This paper applies published methods to a published dataset.

Additionally, gaining a greater understanding of the variables that deep learning models use to predict clinical features from histology images is of critical importance for the translation of such black box models to the clinical setting. Prior work has used autoencoders to extract features from breast cancer histology data, including work from Xu et al (2015, 10.1109/TMI.2015.2458702) and Xie et al (2019, 10.3389/fgene.2019.00080). The use of low dimensional latent representations allow the authors to gain some insight into the variables performant models are utilizing for prediction, like tile hue for 2 latent dimensions. The major advance that this paper offers is a study of how compressing histology images using a convolutional autoencoder effects deep learning classification results.

Major points and limitations:

#While an investigation into the effects of dimensionality reduction could be valuable to better understand prediction-associated features I believe this presented work only superficially explores this topic.

#The data towards assessing heterogeneity is PAM50 subtype tile-level predictions and values for patient-level model accuracy. It is difficult to make any conclusions from predictions of one label shown for 6 slides. While characterizing the heterogeneity of a tumor is valuable and deep learning applied to readily available H&E scanned slides would be an exciting tool for gaining such valuable information, this paper does not generate results towards this end. Ground truth spatial labels would be needed to evaluate whether this work does “support the assessment of heterogeneity” [346].

#The authors note that their motivation for using an autoencoder to reduce the dimensionality of the data is to lessen the compute resources required to train classification models [227]. While it does take substantial compute resources to train large classification models, the most performant models in the field are massive multi-institutional foundation models. Therefore, hospitals implementing classification models for patient care would only be evaluating models for the patient at hand, a relatively trivial computational task. Additionally, the degradation of performance with dimensionality reduction is a significant concern for the clinical setting where the accuracy of tests is paramount. I feel that this is not the proper motivation for the subsequent work.

#Ki67 is the only biomarker with results reported that allow for critical evaluation of how dimensionality reduction impacts performance. The performance for all labels should be reported.

#In Figure 9 it is expected that some information relevant to label classification is lost at the latent dimension is reduced, however, the percentage of instances for which the random inertia is above the label inertia remains relatively stable. From table 3 we can note validation accuracy >0.5, AUROC > 0.5, and F1 score >0.5, meaning that some information is lost but at 2 latent dimensions some relevant information for Ki67 classification remains. A deeper exploration of the information encoded in the latent space that allows for better than random prediction is needed. Examples of supporting data would be dimensionality reduced plots (PCA, UMAP, or tSNE) of the latent space colored by class label side by side with the same scatterplot but with tile images plotted for each point. This should be done for each dimensionality of latent space.

#The data is developed from a multi-centre trial, and interpretation of latent space reduced dimensions could rely on site-specific batch effects (Howard Nature Communications 2021). Please provide baseline model performance for the primary endpoints of interest using relevant meta-data variables as solitary inputs. A model understanding predictive performance of trial site as the only inclusion variable is essential.  

# Many figures are of low quality, and have text which are either unreadable or very low resolution (example: Figure 5). 

Minor points:

# All text should be past tense e.g. [60, 61, 98, 100, 130, 183]

# Clarification of how tiles are generated, whether they are over the whole tissue section, a pathologist-annotated tumor region, and if white-space is excluded is needed [153-155].

#The trait descriptions can be shorted and would benefit from including the assay used to generate the label.

---

## [Author Response · Author response to Decision Letter 1]

15 Dec 2024

Dear Editors of PLOS ONE,

Thank you for reviewing and handling our manuscript titled “LATENT REPRESENTATION OF H&E IMAGES RETAINS BIOLOGICAL INFORMATION IN A BREAST CANCER COHORT”. Below, please find our response to all comments.

Reviewer 1:

Comment:

The authors describe a computational approach to classify regions of Haematoxylin and eosin stained Whole Slide Images of breast cancer with corresponding biological/molecular labels and to use autoencoders to generate a latent (lower dimension) representation of the data, highlighting how this impacts retention of clinically relevant information. The study is relevant and the approach appears to be sound but I suggest that the term "clinical" in the title be replaced as the majority of the data appears to be directly tumour-related (pathological/molecular or biological).

Response:

Thank you for this comment, we followed the suggestion and changed clinical to biological.

Comment:

In areas the language is somewhat unclear e.g. lines 273/274 making it difficult to understand. The paper would benefit from some language revision of the text in areas to improve readability and understanding.

Response:

Indeed, we went through the manuscript again and tried to modify the language. This includes the originally not very clear language in lines 273/274 as pointed out. Thank you for this comment.

Reviewer 2:

Comment:

#While an investigation into the effects of dimensionality reduction could be valuable to better understand prediction-associated features I believe this presented work only superficially explores this topic.

Response:

We agree that this is a first step in this research direction, but still think that the manuscript contains a sufficient volume of information and findings to be of interest to the community

Comment:

#The data towards assessing heterogeneity is PAM50 subtype tile-level predictions and values for patient-level model accuracy. It is difficult to make any conclusions from predictions of one label shown for 6 slides. While characterizing the heterogeneity of a tumor is valuable and deep learning applied to readily available H&E scanned slides would be an exciting tool for gaining such valuable information, this paper does not generate results towards this end. Ground truth spatial labels would be needed to evaluate whether this work does “support the assessment of heterogeneity” [346].

Response:

We thank the reviewer for this comment. In general, the lack of large dataset that can compare h&e to special transcriptomics is currently an obstacle to rigorously testing the potential of inferring the heterogeneity from h&E. That said, we believe that our results do suggest that this is potentially possible in the following sense. First, figure (annotation of pathologist) shows comparison of tile level classification to spatial annotation. Second, it is quite clear that if tile level classification is really accurate, then it can be used to infer heterogeneity. Of course, the latter is true only for heterogeneity based on the predicted traits. For example, in Alona et al [https://www.nature.com/articles/s41598-020-75708-z], the authors show that heterogeneity based on Ki67 and miR17, as inferred from h&e affects prognosis. We hope that future work can further support the validity of this approach, beyond just a few traits.

We changed the text in the discussion to be less general and more accurately reflects this point.

Comment:

#The authors note that their motivation for using an autoencoder to reduce the dimensionality of the data is to lessen the compute resources required to train classification models [227]. While it does take substantial compute resources to train large classification models, the most performant models in the field are massive multi-institutional foundation models. Therefore, hospitals implementing classification models for patient care would only be evaluating models for the patient at hand, a relatively trivial computational task. Additionally, the degradation of performance with dimensionality reduction is a significant concern for the clinical setting where the accuracy of tests is paramount. I feel that this is not the proper motivation for the subsequent work.

Response:

Agreed, we removed reference to computational resources and changed the opening paragraph of the section of autoencoders.

Comment:

#Ki67 is the only biomarker with results reported that allow for critical evaluation of how dimensionality reduction impacts performance. The performance for all labels should be reported.

Response:

We chose to focus on KI67 because, first, it is a very important prognostic marker, and second, literature has already shown its association to morphology. It therefore serves as a baseline case for our investigation. We feel like including the other biomarkers will overload the paper. It also requires more label specific training. A survey of more biomarkers is possibly a good topic for a follow-up paper. We hope the reviewer can agree with this position, if not, we can start the training process for more biomarkers.

Comment:

#In Figure 9 it is expected that some information relevant to label classification is lost at the latent dimension is reduced, however, the percentage of instances for which the random inertia is above the label inertia remains relatively stable. From table 3 we can note validation accuracy >0.5, AUROC > 0.5, and F1 score >0.5, meaning that some information is lost but at 2 latent dimensions some relevant information for Ki67 classification remains. A deeper exploration of the information encoded in the latent space that allows for better than random prediction is needed. Examples of supporting data would be dimensionality reduced plots (PCA, UMAP, or tSNE) of the latent space colored by class label side by side with the same scatterplot but with tile images plotted for each point. This should be done for each dimensionality of latent space.

Response:

Indeed, it's interesting to better understand the behavior in the latent space. We added t-sne figures for Ki67, TILS and PR for different dimensions in the supplement.

Comment:

#The data is developed from a multi-centre trial, and interpretation of latent space reduced dimensions could rely on site-specific batch effects (Howard Nature Communications 2021). Please provide baseline model performance for the primary endpoints of interest using relevant meta-data variables as solitary inputs. A model understanding predictive performance of trial site as the only inclusion variable is essential.

Response:

Of course, in general, the sample origin can be a confounding factor, as pointed out by the reviewer. In our cases however, all patients started as Lum B, and the responsive ones become Lum A. The testing for LumA vs LumB is centrally done in IDIBAPS lab in all three treatment related timepoints. We therefore think that this is not an issue in our case.

Comment:

# Many figures are of low quality and have text which are either unreadable or very low resolution (example: Figure 5).

Response:

We improved the resolution of the figures.

Comment:

# All text should be past tense e.g. [60, 61, 98, 100, 130, 183]

Response:

We reviewed the manuscript and modified the language.

Comment:

# Clarification of how tiles are generated, whether they are over the whole tissue section, a pathologist-annotated tumor region, and if white-space is excluded is needed [153-155].

Response:

This is an important point, thank you for the comment. We explained this better and added an explanatory figure Appendix Figure 2.

Comment:

#The trait descriptions can be shorted and would benefit from including the assay used to generate the label.

Response:

We added description on the assay for each trait and shortened the descriptions for some traits.

Please, let us know if there is anything else we should change.

Thank you very much,

Best regards,

Chloé Benmussa

---

## [Decision Letter · Decision Letter 1]

27 May 2025

PONE-D-24-28161R1Latent representation of H&E images retains biological information in a breast cancer cohortPLOS ONE

Dear Dr. Benmussa,

Thank you for submitting your manuscript to PLOS ONE. After careful consideration, we feel that it has merit but does not fully meet PLOS ONE’s publication criteria as it currently stands. Therefore, we invite you to submit a revised version of the manuscript that addresses the points raised during the review process. Upon review, we noticed that several of the figures appear blurry and low in resolution, which makes it difficult to interpret the data clearly. Could you please replace all figures with high-quality, high-resolution versions (ideally ≥300 dpi) so that labels, axes, and graphical details are crisp and easily readable?We appreciate your attention to this matter and look forward to receiving the revised manuscript with improved figures.

We look forward to receiving your revised manuscript.

Kind regards,

Amgad Muneer

Academic Editor

PLOS ONE

Journal Requirements:

Reviewers' comments:

Reviewer's Responses to Questions

**Comments to the Author**

1. If the authors have adequately addressed your comments raised in a previous round of review and you feel that this manuscript is now acceptable for publication, you may indicate that here to bypass the “Comments to the Author” section, enter your conflict of interest statement in the “Confidential to Editor” section, and submit your "Accept" recommendation.

Reviewer #1: All comments have been addressed

Reviewer #2: All comments have been addressed

2. Is the manuscript technically sound, and do the data support the conclusions?

Reviewer #1: (No Response)

Reviewer #2: Yes

3. Has the statistical analysis been performed appropriately and rigorously? 

Reviewer #1: (No Response)

Reviewer #2: Yes

4. Have the authors made all data underlying the findings in their manuscript fully available?

Reviewer #1: (No Response)

Reviewer #2: Yes

5. Is the manuscript presented in an intelligible fashion and written in standard English?

Reviewer #1: (No Response)

Reviewer #2: Yes

6. Review Comments to the Author

Reviewer #1: (No Response)

Reviewer #2: (No Response)

7. PLOS authors have the option to publish the peer review history of their article (what does this mean?). If published, this will include your full peer review and any attached files.

Reviewer #1: No

Reviewer #2: No

---

## [Author Response · Author response to Decision Letter 2]

10 Jul 2025

Dear Editors of PLOS ONE,

Thank you for reviewing and handling our manuscript titled “LATENT REPRESENTATION OF

H&E IMAGES RETAINS BIOLOGICAL INFORMATION IN A BREAST CANCER COHORT”.

Below, please find our response to the comment.

Comment:

“Upon review, we noticed that several of the figures appear blurry and low in resolution, which

makes it difficult to interpret the data clearly. Could you please replace all figures with high-

quality, high-resolution versions (ideally ≥300 dpi) so that labels, axes, and graphical details are

crisp and easily readable?”

Response:

We recreated all the figures to have a resolution of 300 dpi and checked the dimensions of all the

images to be in the range of the accepted format.

Since there is no change needed in the manuscript, the files ‘Revised Manuscript with Track

Changes’ and the ‘Manuscript’ contain the same file, which is the latest version of the Manuscript

uploaded to the previous submission.

Please, let us know if there is anything else we should change.

Thank you very much,

Best regards,

Chloé Benmussa

---

## [Editor Report · Decision Letter 2]

14 Jul 2025

Latent representation of H&E images retains biological information in a breast cancer cohort

PONE-D-24-28161R2

Dear Dr. Benmussa,

We’re pleased to inform you that your manuscript has been judged scientifically suitable for publication and will be formally accepted for publication once it meets all outstanding technical requirements.

Kind regards,

Amgad Muneer

Academic Editor

PLOS ONE

---

## [Editor Report · Acceptance letter]

PONE-D-24-28161R2

PLOS ONE

Dear Dr. Benmussa,

I'm pleased to inform you that your manuscript has been deemed suitable for publication in PLOS ONE. Congratulations! Your manuscript is now being handed over to our production team.

Kind regards,

on behalf of

Dr. Amgad Muneer

Academic Editor

PLOS ONE